# Anticancer Activity of Natural and Semi-Synthetic Drimane and Coloratane Sesquiterpenoids

**DOI:** 10.3390/molecules27082501

**Published:** 2022-04-13

**Authors:** Lorenz Beckmann, Uta Sandy Tretbar, Reni Kitte, Maik Tretbar

**Affiliations:** 1Medical Faculty, Institute for Drug Discovery, Leipzig University, Bruederstr. 34, 04103 Leipzig, Germany; lb45wyta@studserv.uni-leipzig.de; 2Department of GMP Process Development/ATMP Design, Fraunhofer Institute for Cell Therapy and Immunology IZI, Perlickstr. 1, 04103 Leipzig, Germany; sandy.tretbar@izi.fraunhofer.de (U.S.T.); reni.kitte@izi.fraunhofer.de (R.K.)

**Keywords:** natural products, organic synthesis, anticancer activity, mode-of-action

## Abstract

Drimane and coloratane sesquiterpenoids are present in several plants, microorganisms, and marine life. Because of their cytotoxic activity, these sesquiterpenoids have received increasing attention as a source for new anticancer drugs and pharmacophores. Natural drimanes and coloratanes, as well as their semi-synthetic derivatives, showed promising results against cancer cell lines with in vitro activities in the low micro- and nanomolar range. Despite their high potential as novel anticancer agents, the mode of action and structure–activity relationships of drimanes and coloratanes have not been completely enlightened nor systematically reviewed. Our review aims to give an overview of known structures and derivatizations of this class of sesquiterpenoids, as well as their activity against cancer cells and potential modes-of-action. The cytotoxic activities of about 40 natural and 25 semi-synthetic drimanes and coloratanes are discussed. In addition to that, we give a summary about the clinical significance of drimane and coloratane sesquiterpenoids.

## 1. Introduction

According to the World Health Organization, cancer is the most common cause of death in the 21st century and one of the world’s major health challenges [1,2,3]. Particularly, the growing world population and increasing life expectancy will lead to a constant increase in cancer cases per year. Therefore, the development of effective anticancer treatments is indispensable. Cancer treatment often includes the application of chemotherapeutic drugs, which have become an essential part of cancer drug therapy [4]. Most chemotherapeutic agents are cytotoxic compounds targeting fast-dividing cells by impairing processes of cell division. In general, cytotoxic anticancer drugs include alkylating agents, platinums, anti-metabolites, topoisomerase I and II inhibitors, tubulin-binding drugs, DNA intercalators, and DNA cleaving agents [5,6]. Despite the broad clinical use of chemotherapeutics, there are several drawbacks such as severe side effects due to the lack of selectivity or development of drug resistance [7,8]. Consequently, the investigation of improved anticancer drugs represents a continuous task in medicinal chemistry.

Natural products have proven to be a versatile source of new drugs and pharmacophores in recent decades [9,10,11]. For example, approved chemotherapeutics—such as paclitaxel, vinblastine, or belotecan—are natural products or contain natural product pharmacophores. Despite the long history of natural products as anticancer drugs, their potential has not been exhausted so far. Among natural products, sesquiterpenoids are a prominent group, which has been shown to exhibit cytotoxic effects against cancer cells [12]. Some sesquiterpenoids—such as artemisinin, thapsigargin, or mipsagargin (G-202)—have already entered clinical trials as novel chemotherapeutic agents, showing that this class of natural products holds great potential in anticancer therapy [13]. In nature, sesquiterpenoids are formed in higher plants and microorganisms often to act as antifeedants, deterrents, or attractants [14]. They are 15-carbon compounds derived from the assembly of three isoprenoid units and can be subdivided into acyclic, monocyclic, bicyclic, and tricyclic derivatives. Some examples of cytotoxic sesquiterpenoids are depicted in Figure 1. In this review, we will focus on bicyclic drimane and coloratane sesquiterpenoids.

The general structure of drimanes is derived from drimenol (**4**), which was first isolated in 1948 from the stem bark of *Drimys winterii* (Figure 2) [15]. Drimane sesquiterpenoids are characterized by a trans-decalin core structure and common methyl groups at the C4 and C10 positions [16]. Natural drimanes are often substituted by different oxygen-containing groups—such as aldehyde, hydroxy, acetate, and lactone functionalities—which mainly occur at the C6, C9, C11, and C12 positions [16,17,18,19,20,21]. Rearranged drimanes—also known as coloratanes—are closely related compounds characterized by the 1,2 rearrangement of a C4 methyl group, leading to a terminal double bond at the C4 position.

Sources rich in naturally occurring drimanes and coloratanes are liverworts, ferns and higher plants including *Canellaceae*, *Polygonaceae*, and *Winteraceae* species [22,23]. Due to the pharmacological effects of drimane and coloratanes, some of these plants are commonly used in traditional medicine [24,25,26,27]. Other sources of drimanes and coloratanes are fungi [28,29,30] and marine sponges [31,32,33]. Drimane and coloratane sesquiterpenoids have been shown to exhibit antibacterial [34,35,36], antifungal [37,38], antifeedant [39,40,41], and antimalarial [42,43,44] activity.

This review focuses on the cytotoxic properties of drimane and coloratane sesquiterpenoids against cancer cells. We give an overview of the anticancer activity of natural and semi-synthetic drimanes and coloratanes as well their modes of action. In addition to that, the clinical significance of this class of sesquiterpenoids will be summarized.

## 2. Anticancer Properties

### 2.1. Natural Drimanes and Coloratanes

Natural sources provide structural diverse and bioactive drimane and coloratane sesquiterpenoids. In this subsection, cytotoxic drimanes and coloratanes from natural sources are described and some structure–activity relationships are discussed. All values that are relevant for the discussion of the cytotoxic activity are summarized in Table 1 at the end of this subsection.

#### 2.1.1. Plant Sources

In 1980, Mahmoud et al. showed that natural drimanes from *Cinnamodendron dinisii*—cinnamodial (**7**), capsicodendrin (**8**), and cinnamosmolide (**9**)—exhibit cytotoxic activities against P388 leukemia cells and cells of nasopharynx cancer (Eagle’s 9KB-cells, Figure 3) [45]. It is supposed that the cytotoxic activities of **7** and **8** are similar due to the tendency of the acetal group of **8** to hydrolyze under aqueous conditions to yield **7**. Interestingly, no cytotoxicity was observed for drimanes lacking an α,β-unsaturated double bond at the C7,8 position such as **10**. Karmahapatra et al. also obtained cinnamodial (**7**) and capsicodendrin (**8**) from the species *Cinnamosma fragrans* and tested their cytotoxic and antiproliferative activity against HL-60 and K562 leukemia cells [46]. Cinnamodial (**7**) was less cytotoxic than capsicodendrin (**8**). Despite that, **7** is supposed to be the active species because it was shown that **8** is converted to **7** in DMSO. It is suggested that **8** is a prodrug of **7** and the in situ formation of **7** inside the cell increases its cytotoxic activity.

Nomoto et al. isolated drimane lactones and lactams from *Cinnamosma fragrans* and evaluated their cytotoxicity against A549 carcinoma cells (Figure 4) [47]. Lactone **11** was more potent than **12** and **13**, suggesting that certain hydrophilic substituents could lead to an increase in cytotoxic activity due to improved bioavailability.

Fratoni et al. purified drimanes from the stem bark of *Drimys brasiliensis* and investigated their cytotoxic potential against leukemia cell lines K562 and Nalm-6 (Figure 5) [48]. The drimanes **14**–**19** and **2** were active against both cancer cell lines. Especially, the cinnamoyl derivatives **18** and **19** exhibited increased cytotoxic activities. Compound **18** was 10- to 20-fold more active than polygodial (**2**), showing an increase in cytotoxicity by the conjugated π-substituent. As previously reported, drimanes lacking an α,β-unsaturated double bond at the C7,8 position lead to no or a strongly decreased cytotoxic activity. In further experiments, Fratoni et al. tested derivatives **16**, **18**, and **19** against 16 human cancer cell lines [49]. Compound **18** was the most active compound against most of the cancer cell lines. For seven cancer cell lines, the IC_50_ values of **18** are located within 1–2 µM. Interestingly, there are a few exceptions where **19** was more active than **18**, suggesting a cell-type dependent susceptibility.

Allouche et al. isolated cinnamoyl derivatives **17**, **20**–**22**, and polygodial (**2**) from the bark and leaves of *Zygogynum pancheri* and *Zygogynum acsmithii* and tested their cytotoxic activity against KB adenocarcinoma, HL-60 leukemia, and HCT116 colon cancer cell lines [50]. The cinnamoyl derivatives **17**, **20**–**22** showed increased cytotoxicity compared to polygodial (**2**) as reported by Fratoni et al. [48].

The glycosylated drimane **23** was isolated by Dong et al. from seeds of *Antairis toxicaria* (Figure 6) and exhibited a moderate cytotoxic activity towards K562 leukemia and SMMC-7721 HeLa cell lines [51].

Xu et al. isolated several drimanes from the stem bark of *Warburgia ugandensis* and evaluated their cytotoxicity towards nasopharyngeal cancer cells (KB cells) (Figure 7) [52]. Only the dialdehyde drimanes polygodial (**2**), warburganal (**24**), and mukaadial (**25**) showed a strong cytotoxic activity. Compound **24** was more active than **2** suggesting that the C9 hydroxyl group leads to an increase in cytotoxicity. On the other side, **25** is less cytotoxic than **2** showing that a C6 hydroxylation is disfavored or there is a critical number of hydrophilic groups, leading to a loss of anticancer activity by reduced membrane mobility.

Kitte and Tretbar et al. obtained the sesquiterpenoids polygodial (**2**), warburganal (**24**), and muzigadial (**26**) from the stem bark of *Warburgia ugandensis* and tested their cytotoxic activity against leukemia (MV4-11 and THP-1), melanoma (Sk-Mel29), and glioblastoma (LN-229) cell lines (Figure 7) [53]. Muzigadial (**26**) exhibited three-to-five-fold higher activities than polygodial (**2**). The activity of warburganal (**24**) was also increased compared to polygodial (**2**) as previously observed by Xu et al. [52]. The results by Kitte and Tretbar et al. suggest that a terminal double bond at the C4 position of muzigadial (**26**) increases its cytotoxicity.

#### 2.1.2. Marine Sources

Sakio et al. obtained drimane lactones and acetals **27**–**33** from *Dendrodoris carbunculosa* (Figure 8) [54]. Drimanes **27**–**33** exhibited cytotoxic effects against the lymphoma cell line P388 and adriamycin-, and vincristin-resistant (ADR, VDR) P388 cells with IC_50_ values of 4–17 µg/mL. Lactone **33** showed increased cytotoxicity compared to **27**–**32**. Simultaneous incubation of P388 cells with drimanes **27**–**33** and adriamycin or vincristine led to no reversal effects.

Drimane acetals **34**–**36** were isolated from *Perenniporia maackiae* by J. Kwon et al. and tested against ACHN renal, HCT-15 colon, MDA-MB-231 breast, NCI-H23 lung, NUGC-3 stomach, and PC-3 prostate cancer cell lines (Figure 9) [55]. Compound **35** was shown to be the most active compound. In general, Kwon et al. observed that only drimane acetals with an (*S*)-configuration or a keto group at the C6 position proved to be cytotoxic during their investigations.

#### 2.1.3. Fungal Sources

Liu et al. isolated polyene-substituted drimanes **37**–**39** from the fungus *Aspergillus ustus* and investigated their impact on a murine lymphoma cell line (L5178Y) on PC-12 and HeLa cells (Figure 10) [56]. Compound **37** was shown to be cytotoxic towards all cancer cell lines, whereas **38** and **39** were less active than **37**. In general, L5178 cells were more susceptible towards **37**–**39** than PC-12 and HeLa cells.

Lu et al. also obtained the polyene-substituted drimanes **37** and **38** and an additional compound **40** from *Aspergillus ustus* and investigated their cytotoxicity against A549 carcinoma and HL-60 leukemia cell lines (Figure 10) [57]. The highest activities were observed for **37** and **40**. Surprisingly, **37** exhibited no cytotoxicity against A549 cells and **40** no activity against HL-60 cells. For **38** only a minor impact against A549- and HL-60 cells was observed. The results by Liu et al. and Lu et al. suggest that hydrophilic and electron-deficient groups at the terminal position of the conjugated π-system (Figure 10, green circle) are beneficial for an increase in cytotoxic activity [56,57].

Liu et al. isolated dihydroxylated polyene-substituted drimane lactones **41**–**44** from *Aspergillus flavus* (Figure 11) [58]. All sesquiterpenoids **38**–**41** displayed IC_50_ values ranging from 1.4 to 8.3 µM against HeLa, MCF-7 breast cancer, MGC-803, and A549 cancer cells, of which **41** exhibited the strongest cytotoxicity.

W. Feng et al. used *Fomes officinalis* as a source of drimane sesquiterpenoids (Figure 12) [59]. Only drimane **45** exhibited a moderate cytotoxic activity against HL-60 leukemia, Bel-7402 hepatoma, and KB nasopharyngeal cells.

S. Ngokpol et al. isolated drimane sesquiterpene-conjugated amino acids **46**–**48** from *Talaromyces minioluteus* and evaluated their cytotoxicity against the human hepatocellular carcinoma cell line HepG2 (Figure 13) [60]. Compounds **47** and **48** are four-fold more active than **46**.

All cytotoxic activities of drimane and coloratane sesquiterpenoids from this subsection are concluded in the following table (Table 1). The compounds are classified by their source and the anticancer activity of each compound refers to certain cell lines.

**Table 1 molecules-27-02501-t001:** Anticancer activity of natural drimane and coloratane sesquiterpenoids ^1^.

Source	Species	Comp.	Anticancer Activity ^2^	Value (Unit) ^3^	Cell Lines	Ref.
Plants	*A. toxicaria*	23	6.60 a, 7.20 b	IC_50_ (μg/mL)	a: K562, b: SMMC-7721	[51]
	*C. dinisii*	7	2.2	EC_50_ (μg/mL)	P388 (9: KB)	[45]
		8	2.9			
		9	1.2			
	*C. fragrans*	7	0.48 a, 0.21 b	EC_50_ (µM)	a: HL-60, b: K562	[46]
		8	0.20 a, 0.09 b			
	*C. fragrans*	11	44.6	IC_50_ (µM)	A549	[47]
		12	62.0			
		13	69.6			
	*D. brasiliens*	2	78.20 a, 83.85 b	IC_50_ (µM)	a: Nalm6, b: K562	[48] ^4^
		14	68.03 a, 16.24 b			
		15	78.67 a, 16.66 b			
		16	23.0 a, 26.10 b			
		18	8.18 a, 3.56 b			
		19	12.73 a, 12.98 b			
	*W. ugandensis*	2	1.0	IC_50_ (µM)	KB	[52]
		24	0.3			
		25	5.3			
	*W. ugandensis*	2	6.91 a, 2.22 b, 7.81 c, 22.40 d	IC_50_ (µM)	a: MV4-11, b: THP-1, c: Sk-Mel29, d: LN-229	[53]
		24	3.20 a, 1.12 b, 4.99 c, 4.11 d			
		26	2.62 a, 0.44 b, 2.46 c, 2.74 d			
	*Z. pancheri*, *Z. acsmithii*	2	1 a, 1.2 b, 0.7 c	IC_50_ (µM)	a: KB, b: HL-60, c: HCT116	[50]
		17	0.5 a, 0.5 b, 0.2 c			
		20	0.4 a, 0.3 b, 0.1 c			
		21	0.4 a, 0.3 b, 0.1 c			
		22	0.4 a, 0.2 b, 0.1 c			
Marine life	*D. carbunculosa*	27	10.5	IC_50_ (µg/mL)	P388	[54]
		28	17			
		29	11.5			
		30	15			
		31	10.8			
		32	10			
		33	3.2			
	*P. maackiae*	34	2.0 a, 1.8 b, 2.3 c, 2.1 d, 2.4 e, 6.0 f	GI_50_ (µM)	a: ACHN, b: HCT-15, c: MDA-MB-231, d: NCI-H23, e: NUGC-3, f: PC-3	[55]
		35	1.2 a, 1.6 b, 1.5 c, 1.4 d, 1.9 e, 2.0 f			
		36	3.6 a, 5.4 b, 4.0 c, 4.3 d, 2.3 e, 4.6 f			
Fungi	*A. flavus*	41	3.6 a, 1.4 b, 6.8 c, 5.0 d	IC_50_ (µM)	a: HeLa, b: MCF-7, c: MGC-803, d: A549	[58]
		42	2.9 a, 5.7 b, 1.9 c, 2.5 d			
		43	8.3 a, 3.1 b, 7.1 c, 6.3 d			
		44	4.2 a, 6.6 b, 2.3 c, 7.5 d			
	*A. ustus*	37	0.6 a, 7.2 b, 5.9 c	EC_50_ (μg/mL)	a: L5178Y, b: PC-12, c: HeLa	[56]
		38	1.9 a, >10 b, 7.5 c			
		39	5.3 a, >10 b, >10 c			
	*A. ustus*	37	>100 a, 9.0 b	IC_50_ (μM)	a: A549, b: HL-60	[57]
		38	10.5 a, >100 b			
		40	30.0 a, 20.6 b			
	*F. officinalis*	45	51.2 a, 88.7 b, 146.0 c	IC_50_ (µM)	a: HL-60, b: Bel-7402, c: KB	[59]
	*T. minioluteus*	46	193.3	IC_50_ (µM)	HepG2	[60]
		47	50.6			
		48	57.0			

^1^ The comparison of anticancer activities is only possible within the same publication because of varying cell viability assays. ^2^ The anticancer activity refers to a specific cell line which is indicated by superscripts (a–f) for each value within one species. ^3^ IC_50_ is the drug concentration causing 50% inhibition of the desired activity and is primarily applied for a specific target. For whole-cell assays, the use of GI_50_ and ED_50_ (or EC_50_) is recommended. GI_50_ is the concentration for 50% of maximal cell proliferation, whereas ED_50_ (or EC_50_) is the dose/concentration causing 50% of a maximum effect for any measured biological activity [61]. ^4^ See also [49] for more results.

### 2.2. Semi-Synthetic Drimane Derivatives

Synthetic modification of natural products provides a valuable strategy for increasing pharmacological properties [9,10,11]. For drimane modification, polygodial (**2**) turned out to be a suitable target because it can be isolated in sufficient quantities from different sources and possesses a less complex structure with reactive aldehyde groups [62,63]. Montenegro et al. isolated polygodial (**2**), its epimer 9-epipolygodial (**49**), and drimenol (**4**) from *Drimys winteri* [64]. Based on polygodial (**2**), three semi-synthetic drimanes were obtained by acetalization and reduction. Modification of drimenol (**4**) yielded two cytotoxic drimane derivatives. Some examples are given in Figure 14.

All derivatives were evaluated against human prostate cell lines DU-145 and PC-3 as well as the breast cancer cell line MCF-7. An increased cytotoxicity was observed for polygodial (**2**) and the semi-synthetic derivatives **50**, **52**, and **53**. These data allow a more systematic insight into the structure–activity relationships of drimanes. The electrophilic position at the C7 position and the configuration of the C9 position are critical for cytotoxic activity. Reduction of both aldehyde groups of polygodial (**2**) leads to complete inactivation of both epimers of **51**. Interestingly, acetalization of the C11 aldehyde group of **2** as to **50** has no impact on cytotoxic activity. However, it can be assumed that the acetal group of **50** is converted to **2** under aqueous conditions. Furthermore, 9-epipolygodial (**49**) showed no cytotoxicity suggesting a critical role in the configuration of the C9 position. Transforming the C7 position of drimenol (**4**) to an electrophilic keto group (**52** and **53**) leads to cytotoxic activity compared to **4**.

Dasari et al. modified polygodial (**2**) to increase its antiproliferative activity against A549-, Sk-Mel28, MCF-7-, U373-, and Hs683 cancer cells [65]. 9-epipolygodial (**49**) was obtained by an epimerization reaction of polygodial (**2**) (Figure 1). Typical GI_50_ values of **49** were at 2–6 μM and are 10-to-20-fold smaller than GI_50_ values of polygodial (**2**) Thus, contradictory to Montenegro et al. (Table 2), 9-epipolygodial (**49**) displayed a high antiproliferative activity [64]. Certainly, the impact of anticancer agents is cell line dependent-but even for the same cancer cell line MCF-7, both groups obtained contrary results [64,65]. The two groups performed different cell viability assays and control culture treatments to measure the cytostatic effects of **49** which might lead to different results. Dasari et al. measured the cell viability as sextuplicate in only one experiment [65]. On the other side, Montenegro et al. performed three independent experiments in triplicates [64]. Discrepancies in the data could therefore be due to the lack of biological replicates. Despite that, 9-epipolygodial (**49**) also showed antiproliferative effects against the multi-resistant uterus cancer cell lines MES-SA and MES-SA/Dx5, which were measured in two independent experiments. The effects are comparable to the chemotherapeutics paclitaxel and vinblastine.

Based on 9-epipolygodial (**49**), Wittig derivative **54** was synthesized leading to reduced cytotoxicity compared to **49** (Figure 2) [65]. By selective reduction of the double bond of polygodial (**2**), derivative **55** was obtained that is in equilibrium with **56**. The reduced derivatives **55**/**56** still showed some antiproliferative activity suggesting that the presence of a dialdehyde structure causes a part of the cytotoxicity. Indeed, Dasari et al. observed a high reactivity of the dialdehyde compounds with primary amines forming unstable pyrrole adducts [65]. The authors suggested that dialdehyde drimanes might react with the ε-amine group of lysine residues of proteins, leading to the alteration of their functions. However, no protein drimane adducts were identified so far.

Dasari et al. proposed that the reversible formation of bis-acetals might lead to a controlled release of reactive dialdehydes inside the cell and an increased cytotoxicity [65]. Therefore, different bis-acetals were formed from polygodial (**2**) (Figure 3). Bis-acetals **57a**, **b**, **62a**, **b**, **63a**, **b** and **58** showed comparable or even enhanced antiproliferative effects compared to polygodial (**2**). A possible reason is also the enhanced membrane mobility of the bis-acetals because of the less polar structure compared to polygodial (**2**).

In further works, Dasari et al. focused on the Wittig derivatization of the C12 aldehyde group of polygodial (**2**) (Figure 4) [66]. The reactivity of the C12 position is increased compared to the C11 position, enabling the selective modification of **2**. Figure 4 represents the C12 Wittig derivatives synthesized. All derivatives were tested against A549-, Sk-Mel28, MCF-7-, U373-, and Hs683 cancer cells. Best results were obtained for derivative **64** against MCF-7 cancer (GI_50_ = 7.0 µM) being 10-fold more active than polygodial (**2**). In general, GI_50_ values of all Wittig derivatives **64**–**71** were about 21–42 μM for A549-, Sk-Mel-28-, U373-, and Hs683 cancer cells. Additionally, it was demonstrated that **64** forms an isolable pyrrole product with benzylamine. No Wittig derivatives of 9-epipolygodial (**49**) were synthesized.

Antiproliferative effects of 9-epipolygodial (**49**) and the Wittig derivative **64** were investigated more intensively [67,68]. In addition to that, the derivative **73** was obtained by a Horner–Wadsworth–Emmons (HWE) reaction with **72** (Figure 5). Because of the basic reaction conditions, **73** is formed as a C9 epimer. HWE derivative **73** showed robust inhibition of LNCaP prostate cancer cell growth (IC_50_ = 5 µM) but does not exhibit significant cytotoxicity towards PC-3, DU-145 prostate cancer cells, and normal prostate cells (RWPE-1). Compound **73** is also antiproliferative towards oral squamous cell carcinoma cell lines Cal27 and HSC3 and HeLa cells. GI_50_ values were observed between 2 and 8 μM. For H460 lung cancer cells, the antiproliferative effect of **73** is smaller with a GI_50_ value of 70 μM.

Recently, Maslivetc et al. synthesized ethylene glycol-linked dimers of polygodial (**2**) (Figure 15) [69]. Antiproliferative effects of **74a** and **74b** were increased compared to polygodial (**2**). Against A549-, SkMel-28-, U373-, MCF-7-, Hs683-, and B16F10 cell lines GI_50_ values between 4 and 14 μM were obtained for **74a**. A longer chain length of the polyethylene linker leads to a decreased cytotoxicity. Consequently, derivative **74c** shows higher GI_50_ values than polygodial (**2**). The increased cytotoxicity of **74a** and **74b** is explained by the ability to crosslink antitumor targets by pyrrole formation with lysine residues. However, no crosslink adducts were identified by Maslivetc et al. [69].

In general, there are certain modifications of polygodial (**2**) leading to a loss or an increase in cytotoxicity (Figure 16). It was observed that the reduction of both aldehyde groups extinguishes cytotoxic activity [64,65]. On the other side, the selective reduction of the C7–C8 double bond only diminishes the cytotoxicity without a complete loss [65]. Current results suggest that the presence of an α,β-unsaturated system at the C7,8 position is a key feature for high cytotoxic activity. The conversion of the C9 aldehyde group by Wittig- and HWE reactions maintains a π-conjugation with an electron-deficient carbonyl function and leads to an increase in cytotoxicity [65,67,68]. It was also shown, that the C9 configuration influences the anticancer activity but contradictory results are currently present [64,65]. Additionally, the bis-acetalization and dimerization of polygodial (**2**) are strategies to increase its cytotoxic effects [65,69].

## 3. Mode-of-Action–Specific Effects or ‘Just’ Detergents?

Despite the known cytotoxic effect of natural and semi-synthetic sesquiterpenoids, their mode of action is not completely enlightened and different targets are discussed in the literature. The following part aims to give a comparative overview of accomplished results regarding the mode of action of drimane and coloratane sesquiterpenoids. In Figure 17, different modes-of-action are summarized and will be discussed in further detail during this section. It should be mentioned that a compound can act by more than one mechanism, which can change even within the same compound class depending on its structural details. Especially, data derived from experiments using plant extracts must be interpreted carefully because observed biological activity cannot be exclusively depicted to drimane and coloratane sesquiterpenoids.

### 3.1. Detergent Effects on Mitochondrial Membrane

Quite early, polygodial (**2**) was described to inhibit the mitochondrial ATP synthesis in fungus and eukaryotic cells [70,71,72]. Because of its amphiphilic nature, polygodial (**2**) is supposed to act as a non-ionic detergent on the mitochondrial membrane (MM), leading to the distortion of the mitochondrial membrane potential (MMP). The loss of the MMP is known to be a common signal for cells to undergo apoptosis [73]. In this context, Montenegro et al. observed higher mitochondrial membrane permeabilities in cancer cells after treatment with polygodial (**2**) and two other drimanes [64]. Additionally, the activity of caspase 3—a main executor of apoptosis—was increased. Despite that, these results should be interpreted carefully because the permeabilization of the MM and loss of MMP are part of several pro-apoptotic signaling cascades and are not restricted to detergent effects on the mitochondrial membrane, exclusively [73]. Therefore, it must be determined if the loss of MMP results from the detergent effects of the sesquiterpenoids on the MM or occurs as a downstream event of another apoptotic pathway. Furthermore, the detergent effect on subcellular and cellular membranes should be examined.

The susceptibility of cellular membranes strongly depends on their lipid composition and varies between cell types [74]. Especially, cancer cells show abnormal lipid metabolism resulting in altered membrane compositions [75]. An interesting target for detergents is the lysosome which got increasing attention as an anticancer target for lysosomotropic detergents [76,77,78]. These agents accumulate in the lysosome and lead to the permeabilization of the lysosomal membrane, which is a signal for apoptosis via the lysosomal death pathway [79]. For the lysosomes of chronic lymphocytic and acute myeloid leukemia cells, altered membrane composition and expression profiles were observed, leading to higher susceptibility against detergents [77,80].

### 3.2. TRPV1 Ion Channel

Polygodial (**2**), its epimer **49** and warburganal (**24**) were found to be agonists of the non-selective ion channel TRPV1 (transient receptor potential vanilloid 1), which is expressed on the cell surface of several cancer cell lines as well as on the mitochondrial membrane of different human cells [81,82,83,84,85,86,87,88,89,90,91]. TRPV1 regulates the influx of Ca^2+^ ions into the cytosol and plays a role in inflammation, proliferation, and pain [92,93]. It is known that Ca^2+^ signaling is a critical part of the proliferation and apoptosis resistance of cancer cells [94,95]. On the other side, Ca^2+^ overload can induce apoptosis [96,97]. In the past, different TRPV1 agonists, especially capsaicin, were shown to trigger cell death in TRPV1 expressing cancer cells [98,99,100,101,102,103,104,105]. Based on that, Dasari et al. investigated whether the antitumor effects of different drimanes and derivatives including polygodial (**2**) and its epimer **49** correlate with TRPV1 activity [65,66]. They showed that only polygodial (**2**) and some drimanes with minor cytotoxicity act as agonists on the TRPV1 ion channel, but at concentrations much higher than the GI_50_ values determined. In contrast to previous results, no TRPV1 activity for 9-epipolygodial (**49**) was observed. 9-epipolygodial (**49**) shows high antiproliferative effects against several cancer cell lines. In addition to that, no TRPV1 activity was observed for the highly active Wittig derivative **64** and HWE derivative **73**. To support their results, Dasari et al. used computer models docking the drimane compounds into the VBS (vanilloid binding site) of TRPV1, where only for polygodial (**2**) and some inactive compounds energetically relevant docking positions were obtained [65,66]. These results indicate that the anticancer effects of drimanes might be independent of the TRPV1 ion channel.

### 3.3. TRPA1 Ion Channel

Another potential target of drimanes and coloratanes is the TRPA1 (transient receptor potential ankyrin 1) ion channel. TRPA1 is related to TRPV1 and leads to the influx of Ca^2+^ ions into the cytosol [106,107]. It is overexpressed in different cancer cell lines and attracts increasing attention in cancer-related processes [108,109]. For several drimanes and coloratane dialdehyde compounds, the activation of TRPA1 was observed [110,111]. TRPA1 is activated by electrophilic pungent electrophiles, such as isothiocyanates, forming reversible covalent bonds to cysteine residues of the ion channel protein [112,113,114]. Especially, cysteine 621 had been suggested to be most critical for electrophile-binding [115]. For polygodial (**2**), chemical reactions with cysteine residues have been proposed but no defined reaction product could be identified [116,117,118]. Besides cysteine 621, lysine 620 is suggested to be another critical residue for TRPA1 activation. Mathie et al. showed that polygodial (**2**) reacts rapidly with the amino group of Lys-*Nα*-Ac while no reaction occurred with Cys-*Nα*-Ac [111]. In a mutational study, the dialdehydes polygodial (**2**) and isovelleral retained full activity on a triple cysteine to lysine TRPA1 mutant [110]. These results suggest that TRPA1 activation of drimane dialdehyde compounds is mediated by covalent modification of critical lysine residues within TRPA1. However, the contribution of TRPA1 activation to the anticancer activity of drimane and coloratane sesquiterpenoids needs to be evaluated in greater detail.

### 3.4. ROS Generation and DNA Damage

DNA damage results in the activation of a variety of downstream pathways, which induce either DNA repair mechanisms or cell cycle arrest and apoptosis [119]. Especially, DNA double-strand breaks (DSBs) are critical events leading to the activation of apoptotic downstream events. Typical sources for DNA damage and DSBs are methylating agents and oxidative stress by reactive oxygen species (ROS). In terms of anticancer treatment, radiation therapy (RT) is closely related to ROS and DNA damage [120]. Due to ionizing radiation (IR), ROS are generated at localized regions in the nucleus leading to DNA lesions and consequent cell death. The generation of ROS is not restricted to IR and can also result from endogenous (e.g., mitochondria) and exogenous sources (e.g., heavy metals, xenobiotics) [121,122,123]. An overload of ROS induces apoptosis and cell cycle arrest in cancer cells indicating the potential of ROS in anticancer therapy [124].

For the measurement of DSBs, the amount of phosphorylated histone 2AX (γH2AX) has proven to be a valuable marker for DSBs [125]. DSBs cause the phosphorylation of neighboring γH2AX which is assumed to be functional for DNA repair by causing the chromatin to be more accessible for DNA repair [126]. However, there is also evidence that γHA2X is required for DNA ladder formation during apoptosis [127]. Another protein involved in DNA repair is poly-(ADP-ribose)-polymerase 1 (PARP1) [128,129]. During apoptosis, PARP1 is cleaved by caspases 3 and 7, leading to the inactivation of PARP1-mediated DNA repair. Therefore, the concentration of c-PARP1 levels is a marker for apoptosis.

For the Wittig derivative **64** and HWE derivative **73**, increased concentrations of c-PARP1 and γH2AX were observed after exposure to cancer cells [67,68]. Interestingly, the antiproliferative effects of **64** and **73** can be terminated by the addition of the antioxidant N-acetyl cysteine (NAC), suggesting that ROS play a critical role in the mode of action of drimanes by inducing DNA damage. Increased ROS and γH2AX levels were also observed in HT-29 and HCT116 and A549 cells after incubation with extracts of *W. ugandensis* [130,131]. In addition to that, extracts of *W. ugandensis* lead to an increase in c-PARP1 concentrations.

Contradictory to the previous results, Karmahapatra et al. detected no ROS increase in HL-60 leukemia cells after incubation with capsicodendrin (**8**) [46]. Capsicodendrin (**8**) forms cinnamodial (**7**) under aqueous conditions and it was shown that **7** forms covalent bonds with cysteine residues of glutathione—an intracellular thiol-based antioxidant—without altering intracellular ROS concentrations. Despite that, DNA damage still occurred in HL-60 cells after exposure to cinnamodial (**7**). It is suggested that DNA damage might be caused by the 2-alkenal motif of **7**. In a previous study, it was shown that 2-alkenals lead to DNA damage in V79 lung fibroblasts and Caco-2 colorectal carcinoma cells [132].

In general, there are still open questions regarding the generation and contribution of ROS towards the mode-of-action of drimane and coloratane sesquiterpenoids. It needs to be evaluated whether the generation of ROS depends on specific structural details of the sesquiterpenoids and if ROS production is cell-type dependent. Furthermore, the origin of ROS production must be evaluated in further detail.

It is known that ROS can be produced as a response to mitochondrial stress [133]. As mentioned, drimanes and coloratanes might act as detergents on the MM, leading to mitochondrial stress and ROS generation [70,71,72]. Therefore, ROS could be a link between the detergent effects of drimanes and coloratanes on the MM and DNA damage. It is also possible that mitochondrial-induced ROS are generated by Ca^2+^ influx after activation of TRPV1 or TRPA1. An overload of Ca^2+^ ions is known to induce ROS release from the mitochondrion. Additionally, it was shown that ROS can trigger Ca^2+^ release from the endoplasmic reticulum (ER), resulting in a positive feedback loop with accelerated ROS production [134].

### 3.5. Cell Cycle Arrest

The cell cycle comprises a series of cellular events during the growth and division of cells [135,136]. In general, the cell cycle is divided into different phases (G_0_, G_1_-, S-, G_2_-, and M-phase) that are characterized by the action of specific proteins. Important regulators of the cell cycle are cyclin-dependent kinases (CDKs) that require different cyclins for kinase activity [137].

Zhang et al. observed reduced expression of cyclin D1 and E1 after exposure of A549, HT-29, and HCT 116 cells to the extract of the *W. ugandensis* [130,131]. Cyclin D1 and E1 play an important role in the progression from the G_1_ to the S phase [138,139]. The expression of cyclin D1 is tightly regulated by the PI3K/Akt/GSK3β pathway [140]. After incubation of HT-29 cells with *W. ugandensis* extract, expression levels of PI3K, Akt, *p*-Akt, and *p*-GSK3β were changed leading to the suppression of the PI3K/Akt/GSK3β pathway and reduced amounts of cyclin D1 [131]. In addition to that, Zhang et al. measured an increase in P27 in A549 cells [130]. P27 is a negative regulator of CDKs and is ubiquitinylated by S-phase kinase-associated protein 2 (SKP2), reducing its cellular concentration by proteasomal degradation [141,142,143]. Furthermore, SKP2 is negatively regulated by Forkhead-Box-Protein O3 (FOXO3A) [144,145]. An increased expression of FOXO3A was also observed after cell exposure to extracts of *W. ugandensis* [130]. In future experiments, it would be important to evaluate whether these results are caused by drimane and coloratane sesquiterpenoids or other bioactive compounds from *W. ugandensis*. In this context, Lohberger et al. have already shown that a dehydrocostus lactone sesquiterpenoid leads to cell cycle arrest in soft tissue sarcoma cells by a decrease in CDK2 levels [146]. Therefore, the impairment of CDK-dependent cell cycle progression could be a more common feature of sesquiterpenoids including also drimanes and coloratanes.

### 3.6. Inhibition of DNA-Binding by NF-kB and Stat-3

Felix et al. observed reduced expression of survivin—an antiapoptotic protein critical in apoptosis resistance of cancer cells—in Colo-320 cells after the addition of the drimane lactone **75** (Figure 18) [147].

In further experiments, it was shown that **75** inhibits DNA binding of the transcription factors NF-kB and Stat-3 to the survivin promoter. DNA binding of NF-κB and Stat-3 to the CMV promoter was not impaired. Interestingly, structurally related eudasmane sesquiterpenoids and meroterpenoids lead to the attenuation of NF-κB dependent pathways [148,149]. Tang et al. have shown that upstream of the NF-κB activation the phosphorylation of p38 mitogen-activated protein kinases (p38 MAPK) is inhibited by 1,10-seco-eudesmane sesquiterpenoids [148]. In future studies, it should be evaluated if the inhibition of NF-κB contributes to the anticancer activity of drimane and coloratane sesquiterpenoids.

## 4. Clinical Significance

Around half of the small molecules that have been approved for cancer chemotherapy since 1940 are natural products [150]. All of these compounds were proven to be effective and safe. The efficacy of a compound can only be evaluated if a proper target is identified, which is the main step in drug discovery [151]. Selectivity toward tumor cells preventing off-target effects plays an important role in clinical approval. Although more than 200 compounds derived from natural products are currently in preclinical and clinical investigations [152] only a few of them will be used in clinical routines in the future. The main reason why 95% of small-molecule oncology therapeutics fail in clinical trials is due to a lack of clinical safety and efficacy [153].

Drimane and coloratane sesquiterpenoids were shown to exert promising anticancer activity in vitro. However, no preclinical studies evaluating their in vivo cytotoxic activity nor clinical trials were accomplished until now. In vivo studies are only available for antidiabetic and antimalarial activities of some drimane sesquiterpenoids and plant extracts [44,154]. In these experiments, no unspecific toxicity was observed in rats, indicating a low risk of severe side effects. In addition to that, plant extracts containing drimanes and coloratanes are already used in traditional medicine leading to the conclusion that a clinical application appears feasible [24,25,26,27]. Other sesquiterpenoids, such as sesquiterpene lactones, already made it into clinical trials as chemotherapeutics for their anticancer activity such as artemisinin, showing that sesquiterpenoids are indeed a valuable source for anticancer agents and provide candidates for in vivo studies and clinical trials [13]. We must then ask: why have none of the aforementioned drimanes or coloratanes made it into the clinic yet?

Possible reasons are the lack of detailed knowledge about a specific target or modes-of-action, cancer cell specificity, or insufficient anticancer activities of drimane and coloratane sesquiterpenoids. Until now, IC_50_ values in the low nanomolar range were only observed for a few drimanes and coloratanes against certain cancer cell lines [46,49,50,52,53]. Therefore, the optimization of the anticancer activity of drimane and coloratane sesquiterpenoids should be addressed in future projects. Especially, new synthetic methods could provide access to novel drimanes and coloratanes with enhanced cytotoxicity and selectivity [155,156,157]. Furthermore, a detailed understanding of the mode-of-action would be beneficial for the rational functionalization of the drimane and coloratane scaffold.

## 5. Conclusions

In this review, we gave an overview of the cytotoxic activity of drimane and coloratane sesquiterpenoids against cancer cells. Over the past decades, natural drimanes and coloratanes were isolated and their cytotoxic potential was evaluated. Additionally, several semi-synthetic derivatives were obtained from polygodial to increase its cytotoxic activity. It is demonstrated that the drimane and coloratane scaffold holds great potential as a lead structure for anticancer drug development. However, none of the components are used as a therapeutic agent or in a clinical trial so far. Furthermore, several modes-of-action of drimane and coloratane sesquiterpenoids were discussed. Detergent effects on subcellular membranes, DNA damage, and ROS production are commonly observed for drimanes. It was also shown that cancer cell exposure to drimane-containing plant extracts can cause cell cycle arrest. Additionally, dialdehyde drimanes activate the TRPA1 ion channel. The related ion channel TRPV1 was often supposed to be addressed by drimanes, but current data show that TRPV1 only plays a minor role in the cytotoxic action of this class of sesquiterpenoids. However, whether there is a relation between these actions or they are caused independently needs to be further evaluated. Therefore, a detailed analysis of differentially expressed genes using microarrays or next-generation sequencing methods could be key methods to reveal targets or mode-of-action of drimane and coloratane type sesquiterpenoids in anticancer activity. Proper target identification will lay a solid basis for future lead compound discovery and candidate selection for preclinical and clinical development.

## Data Availability

Not applicable.

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
