# Peer review of "Anticancer Activity of Natural and Semi-Synthetic Drimane and Coloratane Sesquiterpenoids"

_molecules, 2022, doi:10.3390/molecules27082501_

Round 1

Reviewer 1 Report

The author summarized  the cytotoxic properties of drimane and coloratane sesquiterpenoids against cancer cells. ِAlso give an overview of the anticancer activity of natural and semi-synthetic drimanes and coloratanes as well their modes-of-action. The manuscript is well written. Authors listed quite many applications on drimane and coloratane sesquiterpenoids against cancer cells.

  • Line 17, our review not our article.
  • In abstract I think you need to add some data from the review.
  • Table 1 not mention in the text or written as Tab 1.
  • In table 1, you may divide the creatures that produce sesquiterpenoids into three distinct categories: plants, marine, and fungi. Much more clearly, I believe.
  • Table 1 need to be optimized and compacted.
  • Line 317: If summerize the Mode-of-Action in one table or colored figure, I think much better.
  • Clinical Significance,What's the point of including it? The aim or review make no mention of it. The medicinal importance of drimane and coloratane sesquiterpenoids should be included in the table of figures, if you choose to keep it in the article.

Author Response

Dear Reviewer 1,

we thank you for your comments. Please see the attachment for our responses.

Sincerely,

Lorenz Beckmann

Reviewer 2 Report

The authors did a great job reviewing this challenging topic. I would be very happy to see this review in press soon. 

However I have some remarks, it would be great if the authors consider them before publication:

  1. English language polishing is always a plus :-)
  2. Section 4. Clinical Significance is very brief, and I would encourage the authors to extend it a bit. 
  3. I highly recommend the authors to include the publication from tang et al doi.org/10.1016/j.ejmech.2021.113713 to their NFKB section, it presents very interesting findings regarding TLR4/NF-κB/MAPK pathways.
  4. I also highly recommend the authors to include the publication from Sterner and Aperendero et al https://doi.org/10.1021/np0600954  , it presents very interesting findings regarding drimanes as  new anti-inflammatory chemotypes through NFKB blockage.
  5.  I also highly recommend the authors to include the publication from Lohberger et al doi.org/10.1371/journal.pone.0066300 , it presents very interesting findings regarding sesquiterpenoids modulating cell cycle  in soft tissue sarcomas. 

Thank you very much

Best Wishes

Author Response

Dear Reviewer 2,

we thank you for your comments. Please see the attachment for our responses.

Sincerely,

Lorenz Beckmann
